# The Contribution of Cognitive Factors to Compulsive Buying Behaviour: Insights from Shopping Habit Changes during the COVID-19 Pandemic

**DOI:** 10.3390/bs12080260

**Published:** 2022-07-29

**Authors:** Raffaella Nori, Micaela Maria Zucchelli, Laura Piccardi, Massimiliano Palmiero, Alessia Bocchi, Paola Guariglia

**Affiliations:** 1Department of Psychology, University of Bologna, 40127 Bologna, Italy; micaela.zucchelli3@unibo.it; 2Department of Psychology, Sapienza University of Rome, 00185 Rome, Italy; laura.piccardi@uniroma1.it (L.P.); alessia.bocchi@gmail.com (A.B.); 3IRCCS San Raffaele, 00163 Rome, Italy; 4Department of Biotechnological and Applied Clinical Sciences, L’Aquila University, 67100 L’Aquila, Italy; massimiliano.palmiero@univaq.it; 5Department of Human and Society Sciences, University of Enna “Kore”, 94100 Enna, Italy; paola.guariglia@unikore.it

**Keywords:** SARS-CoV-2, working memory, decision-making style, compulsive buying, central executive

## Abstract

The last decade has seen an increase in compulsive behaviours among young adults worldwide, particularly in 2020, during restrictions due to the COVID-19 pandemic. Importantly, even if shopping is an ordinary activity in everyday life, it can become a compulsive behaviour for certain individuals. The aim of this study was to investigate the role of working memory and decision-making style in compulsive behaviour. A total of 105 participants (65 F, 40 M) were recruited online from May 2020 to December 2020. They completed a series of questionnaires to measure shopping compulsive behaviour, decision-making styles, deficits in working memory and online shopping habits. The results show that during the COVID-19 pandemic, people spent much more time shopping online, made more purchases and spent more money than prior to the pandemic. Moreover, both higher working memory deficits and spontaneous decision-making style predicted a greater tendency to engage in compulsive buying. These results suggest the need to develop specific training programs to improve cognitive aspects related to compulsive shopping behaviour.

## 1. Introduction

### 1.1. Background

During COVID-19 [1], global e-commerce grew 58% year-on-year versus 17% in the first quarter of 2020 [2]. Worldwide, shopping site traffic increased 27% year-on-year (28% on personal computers and 29% on mobile devices). Data from the first quarter of 2021 revealed that the spending habits formed during 2020 are destined to be sustained [2] as shoppers capitalize on the large number of everyday opportunities for shopping online. In fact, to prevent the pandemic’s spread, millions of people interrupted their daily activity pattern or transformed their lives into the so-called smart working mode, avoided social interactions and remained isolated at home [3,4]. Even if these measures helped reduce the pandemic’s spread, they also led to a series of negative effects on the general public’s mental health, such as an increase in time spent online [1,5,6].

Although shopping is a routine activity in everyday life, in specific situations it can become a compulsive behaviour for certain individuals. Compulsive buying (CB), originally termed oniomania, is characterized by a concern with shopping and spending that leads to subjective distress and/or impairs quality of life [7,8]. A recent meta-analysis, including 32,000 participants from 16 different countries [9], reported an estimated prevalence of CB of 4.9% in the world population (with very high peaks in several countries, for example 11.3% in Italy; [10]), with higher estimates among university students (8.3%) and customers of a shopping mall (16.2%). These data reflect pre-COVID-19 circumstances and because we do not know whether the increase in time spent online also had an effect on CB, this prompted us to investigate. In fact, this study aimed to assess changes in individuals’ online shopping behaviours both prior to and during pandemic restrictions.

CB is not included in the fifth edition of the Diagnostic and Statistical Manual of Mental Disorders [11] because its diagnostic criteria are insufficiently established. However, most experts believe that CB is best understood within a behavioural addiction framework [12]. Similar to individuals with substance-related disorders, certain compulsive buyers have reported overpowering urges to buy, repetitive loss of control, overspending, feelings of being “high” when shopping, and a negative emotional state that emerges when they are not shopping. Such factors resemble craving, drug-seeking behaviour and withdrawal symptoms, which characterize substance use disorders [13]. Other shared features include a concern with the behaviour and repeated unsuccessful attempts to cut down or stop the behaviour [14,15].

The Interaction of Person-Affect-Cognition-Execution (I-PACE; [16,17]) model can be used as a theoretical framework to investigate CB. This model describes the addictive use of internet applications, such as gambling, gaming, social networking, cybersex and buying/shopping behaviour. According to the model, addictive behaviour is the consequence of interaction among predisposing variables, affective and cognitive responses to specific stimuli and executive functions (i.e., inhibitory control and decision making). P represents the person’s predisposing variables, which may contribute to all types of addictive behaviour [17,18,19]. As noted by Brand et al. [16,17], the decision to behave in a specific manner may be guided by two factors: an impulsive system based on classical and operant conditioning (associative learning) and a reflective system associated with reasoning and executive functions (e.g., [20]). In people with addictions, behaviour is predominantly characterized by the impulsive/reactive nervous system [21], whereas inhibitory control related to the prefrontal cortex may decrease during the addiction process [22,23]. The interaction among these elements is triggered by external cues [17]. External factors specifically related to CB are related to shopping and the marketing environment, such as store size (e.g., [18,24]), ambience (e.g., [25,26]), sales promotions [27,28], whereas for e-commerce they are related to site visual appeal (e.g., [29,30]), navigability [29], security display [29], ease of use [31] and feedback systems [19].

In this study, we focused primarily on the affective and cognitive components of the I-PACE model to better understand the mechanisms underlying CB.

### 1.2. Review Section

Prior research has focused on a variety of personality, demographic and sociocultural factors involved in CB (see [32] for a review), such as impulsive tendency [33,34], self-esteem [35], self-identity and self-image [33], upwards social comparison [36], long-term orientation [37], aspects of the Big Five personality factors [38], negative feelings or moods [39], gender [40], and friendliness [41]. In particular, depression is one of the most prevalent comorbid disorders in individuals suffering from compulsive buying [42,43]. Müller et al. [44] found a significant positive correlation between compulsive buying and depression, which may be associated with feelings of guilt after buying, thus making hopelessness more likely to emerge as a consequence of compulsive shopping [45]. For this reason, we checked for depression in our sample to establish independence between affective and cognitive mechanisms.

Executive functions were also found to play a key role in CB. For example, Lindheimer et al. [46] showed that a general cognitive deficit to resist interference from task-irrelevant external stimuli, measured by the inhibitory motor task (Stroop Matching Task), characterizes individuals with high values in CB screening. Heffernan et al. [47] showed that CB was also associated with impaired self-reported inhibitory control. These results were confirmed in other addictive behaviours, such as compulsive internet gaming and substance dependency [48]. Interestingly, working memory training was found to reduce people’s tendency to act rashly [49]. This means that inhibitory control and working memory can play a key role in CB. In this study, the relationship between CB and working memory deficits, defined in terms of storage, attention and executive function, were assessed in pre and during pandemic buying habits. In addition, given that CB also relies on decision-making processes (see [50]), the role of decision-making styles was investigated. Scott and Bruce [51] defined 5 decision-making styles: the rational style, characterized by searching for information and by logical evaluation of alternatives; the intuitive style, defined by attention to detail and a tendency to rely on intuition and feeling; the dependent style, characterized by searching for advice and guidance from others; the avoidant style, defined by decision-making procrastination; and the spontaneous style, characterized by the tendency to make decisions in an impulsive way and a need to end the decision-making process as quickly as possible. Notably, Gambetti and Giusberti [52] showed that rational and avoidant decision-making styles mediate the influence of self-control and anxiety, respectively, on the decision to spend. Other studies have noted that the individuals abiding by the rational style exhibited a tendency to analyse all available financial, economic and environmental information before deciding to invest, whereas those abiding by the spontaneous style, relying on emotion, made shortcuts rather than carrying out fundamental analysis [53,54,55]. Because of this reduced ability to plan and control decision making, the spontaneous style was often associated with lower managerial effectiveness, reduced propensity to accommodate conflict, lower course percentages in groups of management undergraduates (i.e., [56,57]) and greater unfavourable financial decision making (e.g., gambling; [58]). Moreover, even if intuitive decision making is usually correlated with the spontaneous style [51], the intuitive style has been associated with better performance in decision making; in fact, the latter is characterized by quick processing of information but also with the ability to grasp significant details in the context, and it has been associated with superiors’ ratings of innovativeness [51]. In this study, to understand whether decision-making style has an effect on affective and cognitive mechanisms underlying CB, we also investigated the individual’s propensity to make decisions.

### 1.3. Rationale and Hypotheses

This study examined new factors that can be involved in consumer CB. The main objective of the study was to investigate whether government restrictions aimed to contain the SARS-CoV-2 pandemic and subsequent social isolation resulted in a change in online shopping habits associated with specific cognitive functions, such as working memory capacity and decision-making style, while controlling for demographic characteristics and depression (the latter because of its high association with compulsive buying). For example, exposure to highly stressful events seems to hinder the formation of explicit memories and, more generally, memories based on flexible and complex reasoning [59]. Therefore, we expected such memories to be particularly affected during the isolation period. In particular, regarding CB, changes in the number of purchases, the frequency of shopping per week and the amount of money spent were taken into consideration to explore which of these aspects were most influenced by the spontaneous decision-making style and working memory deficits. Furthermore, we believe that the analysis of the role of these specific cognitive processes, i.e., spontaneous decision-making style and working memory, is relevant not only to understanding current restrictions and social isolation due to the COVID-19 pandemic, but also to predicting CB in daily life.

The research hypotheses were as follows.

**H1.** 
*Pandemic social isolation changed people’s online shopping habits, resulting in an increase in the number of purchases and the frequency of shopping (i.e., the number of times one shopped per week) and in the amount of money spent during pandemic isolation.*


**H2.** 
*Higher levels of working memory (storage, attention and executive function) deficit will be associated with higher CB.*


**H3.** 
*A spontaneous decision-making style will be associated with a greater tendency to CB.*


**H4.** 
*Pandemic social isolation changed online shopping habits of people, characterized in particular by working memory deficits and a spontaneous decision-making style, resulting in an increase in the number of purchases and the frequency of shopping (i.e., number of times one shopped per week) and the amount of money spent during pandemic isolation.*


## 2. Materials and Methods

### 2.1. Participants

Our sample consisted of 125 volunteer participants. Participants were excluded if they were suffering from a condition that might cause difficulty or distress when completing the experiment tasks, such as neurological or psychiatric disorders. Twenty participants were excluded because they did not complete the General Decision-Making Styles questionnaire. The final sample consisted of 105 participants (demographic information is reported in Table 1). To determine the sample size, a power calculation was performed using G*Power 3.1 [60]. To perform the multiple regression analysis (considering two predictors: working memory deficit and spontaneous decision-making style), the following parameters were used: effect sizef2 = 0.15—medium magnitude; alpha = 0.05; power = 0.90. This gave a suggested sample size of at least 88 participants. Participants were recruited online from May 2020 to December 2020 through notices posted on social networks and researcher bulletin boards using Qualtrics software [61]. Of the 105 participants, 83 exhibited minimal depression (0–13 points on the Beck Depression Inventory II), 12 exhibited mild depression (14–19 points on the BDI II scale), 4 exhibited moderate depression (20–28 points on the BDI II scale), and 6 exhibited severe depression (29–63 points on the BDI II scale). Participants also responded to questions regarding their financial income: 48% of the sample had a monthly salary of fewer than 1000 euros; 31% had a monthly salary between 1000 and 2000 euros; 14% had a monthly salary between 2000 and 3000 euros; 6% had a monthly salary of more than 3000 euros. All participants signed a written consent form before the study began. The study was approved by the local ethics committee (Prot. n. 36252, University of Bologna, Italy).

### 2.2. Materials and Procedure

A personal data questionnaire was administered to establish demographic information and to investigate other areas of interest. The demographic information included age, gender, education, financial income, neurological or psychiatric illness, drug use and the use of alcohol or medications. Questions regarding COVID-19 were asked to determine whether the participants had contracted the disease and reported disease-related issues. In addition, a questionnaire regarding online buying habits before and during the COVID-19 pandemic was administered. The latter consisted of a series of questions, which were repeated in the same form but considered two different periods: before the COVID-19 pandemic (March 2020) and during the COVID-19 pandemic (Table 2).

The General Decision-Making Styles (GDMS; [51]; Italian version: [62]) is a self-administered questionnaire designed to assess how individuals approach decision-making situations. It consists of 25 items and the following five scales: rational (thoroughly searching for and logically evaluating alternatives), intuitive (relying on hunches and feelings), dependent (searching for advice and soliciting direction from others), avoidant (postponing and avoiding decisions), and spontaneous (responding to a sense of immediacy and a desire to complete the decision-making process as soon as possible). The 25 items were presented to the respondents in the form of a five-point Likert scale ranging from strongly disagree (1) to strongly agree (5). In prior studies, the GDMS showed adequate internal consistency reliability (Cronbach’s alpha ranged from 0.65 to 0.85 for the rational scale, 0.78–0.84 for the intuitive scale, 0.62–0.86 for the dependent scale, 0.78–0.94 for the avoidant scale and 0.77–0.87 for the spontaneous scale [51,57,63]). Cronbach’s alpha for the present sample was 0.81 for the rational scale, 0.87 for the dependent scale, 0.85 for the spontaneous scale, 0.86 for the avoidant scale and 0.78 for the intuitive scale.

The Working Memory Questionnaire (WMQ; [64], Italian version [65]) is a scale designed to assess working memory deficits in everyday life. It is a self-administered scale, composed by 30 questions in three different domains, each addressing a different dimension of working memory: short-term storage, attention, and executive control. Each question is rated on a five-point Likert-type scale, ranging from 0 (“no problem at all”) to 4 (“very severe problem in everyday life”). Three sub-scores are computed (maximal score 40 for each subscale) as well as a total score (out of 120). Higher scores correspond to more difficulties/complaints. The short-term storage domain corresponds to the ability to maintain information in short-term memory for a short period of time but also investigates mental calculation and written text comprehension. The attention domain assesses distractibility, mental slowness, mental fatigue, and dual-task processing. The third domain is related to executive aspects of working memory, such as decision making, planning ahead, or shifting. In the following analyses, both the total score on the questionnaire and the scores on the three subscales will be considered. In prior studies, the questionnaire revealed a good internal consistency, both for healthy participants and patients with brain injury (Cronbach’s alpha = 0.89 and 0.94, respectively). Moreover, the WMQ has the sensitivity to discriminate patients from matched controls in the three domains (*p* < 0.0001). Good concurrent validity was found for the Cognitive Failure Questionnaire (CFQ; [66]) and the Rating Scale of Attentional Behaviour (RSAB; [67]) (Spearman’s Rho = 0.90 and 0.81, respectively; both *p*s < 0.0001). The total complaint score was significantly correlated with neuropsychological measures of working memory (visual spans and short-term memory with interference) and global intellectual efficiency (Raven’s Matrices) but not with digit spans [64]. Cronbach’s alpha for the present sample was = 0.96.

The Faber and O’Guinn Scale [68]. This scale consists of seven items that describe particular shopping behaviours. Participants must rate how often the item content describes their behaviour using a 5-point Likert-type scale ranging from 1 (very often) to 5 (never). To obtain the total score, a specific calculation provided by the authors should be used. The calculation requires adding all the individual scores obtained in each question, and subtracting 9.69 points. Therefore, the higher the negative score obtained, the higher the tendency of compulsive buying. The Faber and O’Guinn scale is also used for identifying the more extreme cases of CB behaviour [69,70]. Faber and O’Guinn [68] used logistic regression to develop a compulsive buying scale (CBS) to identify individuals suffering from compulsive buying; the CBS correctly classified approximately 88% of the subjects. This scale has become the gold standard in compulsive buying scale research [50]. Cronbach’s alpha for the present sample was = 0.91.

Beck Depression Inventory II (BDI-II [71]). This questionnaire is a 21-question multiple-choice self-report inventory. It is one of the most widely used psychometric tests for measuring the severity of depression. The BDI-II is a 1996 revision of the BDI [72] developed in response to the American Psychiatric Association’s publication of the Diagnostic and Statistical Manual of Mental Disorders, Fourth Edition, which changed many of the diagnostic criteria for major depressive disorder. Each answer is scored on a scale value of 0 to 3, with a range from 0 to 63. Higher total scores indicate more severe depressive symptoms. The standardized cut-offs are as follows: 0–13: minimal depression, 14–19: mild depression, 20–28: moderate depression, 29–63: severe depression. The BDI-II is positively correlated with the Hamilton Depression Rating Scale [73], with a Pearson r of 0.71, indicating good agreement. This test has also been shown to have a high one-week test–retest reliability (Pearson r = 0.93), suggesting that it is not overly sensitive to daily variations in mood. The test also has high internal consistency (alpha = 0.91) [71]. Cronbach’s alpha for the present sample was = 0.93.

All questionnaires were administered online through Qualtrics software [61]. Before starting, participants provided their informed consent. Then, they provided demographic information and completed the questionnaire regarding online buying habits before and during the COVID-19 pandemic. Last, in a randomized order, they completed the BDI-II, Faber and O’Guinn Scale, WMQ, and GDMS. The experimental procedure lasted approximately 20 min.

### 2.3. Statistical Analyses

A series of repeated ANOVAs were performed by considering the answers provided to the questionnaire regarding buying habits before (March 2020) and during (May–December 2020) the COVID-19 pandemic. In addition, a series of hierarchical regressions were used to assess the contributions of demographic characteristics, depression, decision-making styles, and working memory deficit to online shopping behaviours. Data were analysed using the SPSS 26.0 package. The data presented in this study are openly available in the open science framework repository at https://osf.io/gntw3/?view_only=d9eeecc143b14d5c869bbb7df9789971 accessed on 9 June 2022.

## 3. Results

This study aimed to investigate whether changes in online shopping habits, occurring prior to and during the COVID-19 isolation, were associated with working memory capacity and decision-making style, while controlling for demographic characteristics and depression.

**H1.** 
*Changes in online buying habits before and during the COVID-19 Pandemic.*


The first repeated ANOVA considered the number of times one shopped online per week before and during the imposition of COVID-19 restrictions. The results indicated a significant increase in the number of times individuals shopped online per week during social restriction (F1,63 = 8.79, *p* < 0.05, n2 = 0.12) (Mean (Sd) before: 1.89 (0.94) online shopping sessions per week; during the COVID-19: 2.32 (1.08) online shopping sessions per week). The second repeated ANOVA considered the number of purchases made before and during the COVID-19 restriction. The results showed a significant increase in the number of purchases made during social restriction (F1,60 = 5.29, *p* < 0.05, n2 = 0.08) (Mean (S.D.) before: 1.92(0.75); during the COVID-19: 2.30(1.20)). The last repeated ANOVA considered the amount of money spent on online shopping before and during the COVID-19 restriction. The results showed a significant increase in the amount of money spent on shopping online during social restriction (F1,66 = 4.08, *p* < 0.05, n2 = 0.05) (Mean (S.D.) before: 2.48(0.85); during the COVID-19: 2.81(1.15)).

**H2.** 
*Higher working memory deficit predicts a greater tendency to CB.*


In the first hierarchical regression analysis, only the third model was significant. Specifically, a higher working memory deficit was related to more compulsive buying (please note that, in this case and in the following analyses, a negative score for compulsive buying means a higher tendency of this behaviour according to the Faber and O’Guinn scale scoring), as well as higher scores on depression and older age. See Table 3 for statistical values.

In the second hierarchical regression analysis, only the third model was significant. Specifically, a higher deficit in the executive component of working memory was related to more compulsive buying, as well as older age. See Table 4 for statistical values.

**H3.** 
*A higher spontaneous decision-making style predicts a greater tendency to compulsive buying.*


In the first hierarchical regression analysis, only the third model was significant. Specifically, only a higher spontaneous decision-making style score was related to more compulsive buying, as well as higher depression scores. See Table 5 for statistical values.

In the second hierarchical regression analysis, we considered both working memory deficit and spontaneous decision-making style, and only the third model was significant. That is, a higher deficit of the executive component of working memory was related to more compulsive buying, as well as older age. See Table 6 for statistical values.

**H4.** 
*Spontaneous decision-making style and working memory deficit increase the number of purchases and the amount of money spent during the COVID-19 pandemic.*


Having found a positive relationship among compulsive buying, working memory deficit and spontaneous decision-making style, we performed a series of Ancovas to determine which aspects of compulsive buying (number of sessions of online shopping per week, number of purchases and amount of money spent) were affected by the COVID-19 pandemic (within variable: before and during the COVID-19 pandemic), after controlling for working memory deficit (total score and subscales) and spontaneous style.

Regarding the number of sessions per week of shopping online, the results showed a significant increase (F1,61 = 8.67, *p* < 0.05, n2 = 0.12) during social restriction (considering the working memory deficit total score) [2.32(1.08)] compared to before the COVID-19 pandemic [1.89(0.94)]. No other results were significant (*p*s from 0.21 to 0.95). The results showed a significant increase in the number of times per week one shopped online (F1,59 = 8.89, *p* < 0.05, n2 = 0.13) during social restriction (also considering the working memory deficit subscales). No other results were significant (*p*s from 0.06 to 0.69).

Regarding the number of purchases, the results showed a significant increase in the number of purchases made during social restriction (F1,58 = 7.54, *p* < 0.05, n2 = 0.11) [2.30(1.20)] compared to before the COVID19 restrictions [1.92(0.75)], a significant effect of spontaneous style on the number of purchases (F1,58 = 11.01, *p* < 0.05, n2 = 0.16) and a significant effect of the working memory deficit total score on the number of purchases (F1,58 = 9.60, *p* < 0.05, n2 = 0.14). Considering the working memory subscales, we found a significant effect of the executive subscale (F1,56 = 4.69, *p* < 0.05, n2 = 0.07) on the number of purchases, as well as of spontaneous style (F1,56 = 4.83, *p* < 0.05, n2 = 0.07). We examined these interactions in greater detail by performing a series of regression analyses between spontaneous style and working memory deficit scores and number of purchases before and during the COVID-19 restriction. It appears that during the COVID-19 restriction, a higher level of spontaneous style predicted a higher number of purchases (F1,67= 13.84, ß = 0.41, adjR = 0.16, *p* < 0.001), as did the working memory deficit total score (F1,67 = 11.13, ß = 0.38, adjR = 0.13, *p* < 0.05) and the executive working memory deficit subscale score (F1,67= 19.44, ß = 0.47, adjR = 0.21, *p* < 0.001).

Regarding the amount of money spent shopping online, the results showed a significant increase in the amount spent during social restriction (F1,64 = 4.62, *p* < 0.05, n2 = 0.06) [2.81(1.15)] compared to before the COVID19 restrictions [2.48(0.85)] and a significant effect of the working memory deficit total score (F1,64 = 7.05, *p* < 0.05, n2 = 0.09). Considering the working memory deficit subscales, we found a significant effect of the deficit on the executive component of working memory (F1,62 = 8.21, *p* < 0.05, n2 = 0.11). Moreover, we performed a series of regression analyses between working memory and the amount of money spent before and during the COVID-19 restrictions. The results showed that during the second period, a higher working memory deficit total score (F1,74= 18.12, ß = 0.44, adjR = 0.18, *p* < 0.001), and specifically the executive working memory deficit subscale (F1,74 = 28.51, ß = 0.53, adjR = 0.27, *p* < 0.001), predicted more money spent.

## 4. Discussion

This study has shown that during the COVID-19 pandemic, shopping habits changed. People spent much more time shopping online both in navigating online sales sites and in pursing and spending money to buy (H1). This result confirmed the increase in e-commerce during the COVID-19 shown by numerous studies on this specific issue (e.g., [74,75]), i.e., because of the ban on leaving the home to shop, online shopping became a first necessity behaviour. Moreover, this study extends previous research on CB, suggesting that beyond the effects of the pandemic there are also cognitive factors that predict buying behaviour: deficits in working memory and the spontaneous decision-making style.

Regarding working memory, our results supported the hypothesis (H2) that the more impaired the working memory (especially the central executive component), the greater the tendency to CB, supporting the I-PACE model [17], which highlights the crucial role of executive functions in the retrieval and organization of cognitive and emotional information from long-term memory. Our results confirmed that CB is related to impairments in working memory, specifically in the executive component of working memory. In fact, the central executive was found to be involved in predicting a higher tendency to CB both with respect to classical screening scales (see [68]) and to questions exploring the number of items bought or the amount of money spent (see questionnaire regarding shopping habits). Therefore, we can imagine a shopper who is limited in the ability to retain information and to perform comparisons among products being actively maintained in memory. The more information that a person tries to consider and compare simultaneously, the more overloaded the person’s capacity becomes. A possible solution to this difficulty could be making decisions as quickly as possible. However, this solution could result in maladaptive behaviour, such as CB. Low scores in working memory seem to occur not only in habitual purchase situations but also in specific situations (i.e., the COVID-19 pandemic period) determined by social isolation and increased levels of depression (e.g., [76]), leading to an increase in purchases and money spent, both indicators of CB (H4) (e.g., [77]). It seems likely that the external problematic situation produces a cognitive load (i.e., a new situation with new problems and information to process), which increases working memory load, thus accentuating dysfunctional behaviours and resulting, in the specific case, in an increase in the number of purchases and money spent.

These results are consistent with neuropsychological theories that emphasize how prefrontal processes act with the intent to keep in mind goals and planned activities in working memory while inhibiting inappropriate influences from subcortical processes [78,79]. Our data are also in agreement with recent research on the relationship between inhibitory control and working memory [80,81]. Thus, when inhibitory control is impaired (the executive system of working memory), inappropriate affective and cognitive responses to environmental shopping stimuli will drive decision making and behavioural processes [17].

The spontaneous style was also found to be a predictor of CB (H3). This style is characterized by a feeling of immediacy and a need to decide as quickly as possible. For example, in economic decision making, individuals who adopt this style make shortcuts and rely on emotions rather than engage in fundamental analysis. For instance, they accept advice from strangers and pay heed to speculation and rumours (e.g., [52,55]). This approach to financial decision making is similar to CB: the individual must immediately choose what to buy. CB is even more pronounced in stressful situations, such as during the COVID-19 pandemic (H4), during which individuals characterized by this decision-making style increased their number of purchases. Accordingly, results of the present study confirmed that the spontaneous decision-making style was a predictor of CB. In this regard, the discouragement of the adoption of this style could be accomplished by the “nudge” approach. Nudges are defined as “liberty-preserving approaches that steer people in particular directions, but that also allow them to choose their own way” [82] and are applied to different areas of people’s lives, including consumer, health, energy and civic behaviour (e.g., [83,84]), but principally economics [85]. There are different types of nudges, such as promoting existing programs; for example, by involving education, health, or finance, to reduce their complexity. A further type of nudge involves informing people that the majority of their peers are engaged in a certain type of behaviour. In fact, emphasizing what most people do usually creates a social norm that individuals tend to follow. Making healthy food more visible to consumers is another example of a nudge based on the increased ease and convenience of product availability. For example, clearly displaying fruit and vegetables in a campus coffee shop increased students’ consumption of the same, while reducing their intake of less healthy foods [86]. Another example is a digital nudge, such as a repeating phone vibration that nudges a user to decrease their online shopping [87]. Such strategies can help people exhibiting higher levels of the spontaneous decision-making style to undertake more thoughtful and reflective decisions. In conclusion, the COVID-19 pandemic certainly increased CB. However, importantly, its effect was enhanced when individuals were characterized by the spontaneous decision-making style and working memory difficulties. This means that the COVID-19 pandemic had an effect on CB under specific conditions. The only direct effect of the pandemic concerned the amount of time per week spent shopping online during social restriction. However, this finding could be simply due to the greater amount of time that individuals had at their disposal as a consequence of social restrictions.

## 5. Conclusions

In summary, the present study adds to the research by investigating the specific role of cognitive factors (i.e., working memory impairments and spontaneous decision-making style) that contribute to CB. There were several key finding:. We found that working memory impairments, and in particular its executive component related to the ability of decision making, planning ahead, or attentional shifting, are predictors of CB, as well as the spontaneous decision-making style. Therefore, to support the decision-making process, future studies should consider the development of working memory training aimed at improving the ability to manage and control items to be processed in solving problems or making decisions. This approach has previously been found useful for other problems, such as gambling addiction and ADHD [81].

This study is not without limitations. While the use of self-report measures can be considered an advantage for ecological issues, it did not enable us to consider the different aspects of executive functions, such as planning, inhibitory strategies, self-monitoring, and cognitive flexibility [88]. Future research should examine these aspects both with specific tasks and self-report measures. Although we are confident regarding the replicability of our findings, one should consider that the WMQ is an instrument that correlates strongly with subjects’ performance in behavioural tests, as demonstrated by Guariglia et al. [63] and earlier by Vallat-Azouvi, Pradat-Diehl and Azouvi [62] in normal and pathological populations. Another limitation of this paper concerns the fact that human behaviour shows great variability [89,90] and for this reason, the larger a sample is, the more representative it is of the behaviour under study. Nevertheless, the number of participants in this study is congruent with the number of independent variables considered to explain CB. Moreover, the COVID-19 is widely known to have disproportionately impacted the working class [91]; this study was less able to survey this social class. In addition, future researchers could seek to extend knowledge regarding the possible detrimental social and economic consequences of CB by exploring and embedding an investigation of long-studied affective components with more recently studied cognitive ones.

## Figures and Tables

**Table 1 behavsci-12-00260-t001:** Sample demographics, mean (SD), separately for males and females.

	Males (N = 40)	Females (N = 65)
**Age**	35.18 (10.69)	36.15 (10.34)
**Education**	14.40 (3.33)	14.54 (3.25)
**BDI II scale**	8.35 (10.91)	9.23 (8.89)

**Table 2 behavsci-12-00260-t002:** Questionnaire regarding shopping habits before/during the COVID-19 pandemic.

Questions	Scoring
Did you use to shop online before/during COVID-19	Yes (1) No (0)
How many times did you shop online per week before/during COVID-19?	0 times per week (1) 1–2 times per week (2) 3–5 times per week (3) 5–10 times per week (4) more than 10 times per week (5)
How many items did you buy online before/during COVID-19?	1 item (1) 1–2 items (2) 3–5 items (3) 5–10 items (4) more than 10 items (5)
How much did you spend shopping online before/during COVID-19?	5–10 euros (1) 15–50 euros (2) 50–10 euros (3) 100–300 euros (4) more than 300 euro (5)

**Table 3 behavsci-12-00260-t003:** Statistical values of the hierarchical regression analysis.

	F (df)	R^2^	Beta	*p*
**STEP 1**				
age, sex, education and financial income	1.07 (4104)	0.04		0.37
**STEP 2**				
depression	2.22 (5104)	0.10		0.057
**STEP 3**				
working memory deficit total score,	10.91 (6104)	0.40	−0.55	**<0.001**
depression,			−0.22	**<0.05**
age			0.22	**<0.05**

**Table 4 behavsci-12-00260-t004:** Statistical values of the hierarchical regression analysis.

	F (df)	R^2^	Beta	*p*
**STEP 1**				
age, sex, education and financial income	1.07 (4104)	0.04		0.37
**STEP 2**				
depression	2.22 (5104)	0.10		0.057
**STEP 3**				
	11.38 (8104)	0.48		**<0.001**
Working memory Storage deficit			0.17	0.33
Working memory Attention deficit			−0.28	0.11
Working memory Executive Function deficit,			−0.54	**<0.001**
age			0.26	**<0.05**

**Table 5 behavsci-12-00260-t005:** Statistical values of the hierarchical regression analysis.

	F (df)	R^2^	Beta	*p*
**STEP 1**				
Age, sex, education and financial income	1.07 (4104)	0.04		0.37
**STEP 2**				
Depression	2.22 (5104)	0.10		0.057
**STEP 3**				
Decision-making styles scores:	2.49 (10,104)	0.21		**0.01**
Dependent			−0.11	0.40
Intuitive			0.25	0.08
Avoidant			−0.06	0.61
Spontaneous			−0.34	**0.01**
Rational,			−0.09	0.44
depression			−0.23	**<0.05**

**Table 6 behavsci-12-00260-t006:** Statistical values of the hierarchical regression analysis.

	F (df)	R^2^	Beta	*p*
**STEP 1**				
Age, sex, education and financial income	1.07 (4104)	0.04		0.37
**STEP 2**				
Depression	2.22 (5104)	0.10		0.057
**STEP 3**				
	11.38 (8104)	0.48		**<0.001**
Working memory Storage deficit			0.17	0.33
Working memory Attention deficit			−0.28	0.11
Working memory Executive Function deficit, age			−0.54 0.26	**<0.001** **<0.05**
**STEP 4**				
Spontaneous decision-making style score	10.31 (9104)	0.49	−0.09	0.24

## Data Availability

The data presented in this study are openly available in the open science framework repository at https://osf.io/gntw3/?view_only=d9eeecc143b14d5c869bbb7df9789971 accessed on 9 June 2022.

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
