# Peer review of "The Contribution of Cognitive Factors to Compulsive Buying Behaviour: Insights from Shopping Habit Changes during the COVID-19 Pandemic"

_behavsci, 2022, doi:10.3390/bs12080260_

Round 1
Reviewer 1 Report
The contribute of cognitive factors to understand compulsive buying behavior: Insights from shopping habits changes during Covid19 pandemic reports on participants’ online shopping (compulsive buying) behaviors prior to and during the COVID-19 pandemic in conjunction with individuals’ decision-making styles and working memory characteristics. This is an interesting study with applicability beyond pandemic-associated (highly stressful time) outcomes. There are, however, a significant number of issues that must be addressed before publication.
General:
Please be consistent in the use and spelling of COVID-19. Covid19, Covid, Covid19 pandemic, and coronavirus pandemic are some examples from the manuscript.
Grammar (e.g. word order, verb tense shifts, pluralization) needs considerable attention. I provide a few examples from my discussion of the various sections of the paper below, but they are not an exhaustive list.
The title should be revised: The contribution of cognitive factors to compulsive buying behavior: Insights… or something similar.
Introduction
The first sentence is not needed and the claim of “exponential” growth is not supported by citation. Instead, begin During the COVID-19 pandemic, global e-commerce grew….
Lines 39-43 – This sentence is difficult to follow.
Line 45 – do you mean social interactions?
Line 46- instead of involved perhaps use created, led to, etc.
Line 48 – Restate the hypothesis. Perhaps “The present study aimed to assess changes in individuals’ online shopping behaviors prior to and during pandemic restrictions.”
Line 56 – what is meant by shopping specific samples?
Line 71 – buying/shopping or buying and shopping behavior
Line 87 – by internal factors do you mean affective and cognitive components? If so, please state that way.
Line 92 – Wouldn’t employee friendliness be considered external factor?
Lines 106-108.- move the first sentence to the end of the paragraph above.
Line 156 – A spontaneous decision-making style will be associated with a greater tendency toward CB, or Individuals characterized by a spontaneous decision-making style will show a greater tendency toward CB.
Materials and Methods
2.1 This section needs to be reorganized. Move the number of participants (Our sample was composed…) to the beginning of the paragraph. Include the power calculation information after the age range information. The descriptive information would be easier to follow in a table. It would also be good to see the data separately for males and females.
2.2 Proofread carefully and condense. There are details for these instruments that are repeated. Additionally, tense shifts are particularly problematic in this section.
Line 247 – what is meant by Scores are calculated for each participant with a specific equation? The sentence following this sentence also does not make sense.
Line 261 – sentence fragment
Results
This first section should be referred to as Statistical Analyses and may be better suited to the previous section (Materials and Methods). Either use a subheading as you do in Section 3.1 or restate your variables instead of referring to H1, H2, etc. For example, In order to assess changes in individuals’ online shopping habits before and during the pandemic, series of repeated measures ANOVAs (although I am unclear why a paired-samples T-test wasn’t used since there are only 2 time points) were performed. Another possibility is to state your statistical approach more broadly, i.e. hierarchical regression was used to assess the contributions of demographic characteristics, depression, decision-making styles, and working memory on online shopping behaviors, etc.
There is also a lot of repetition in the models (both here and in the actual presentation of the results). Perhaps a table would be better?
If you move the Statistical Analyses to the Methods section, then the Results section should begin with a statement about the overall purpose of the paper. You can then work through the results for each of the hypotheses. Again, consider presenting the data from the regression models in a table. There is so much repetition in this section.
Discussion
This section needs to be carefully proofread. Some sentences (such as lines 432-437) ramble and are difficult to follow. Additionally, the discussion of nudging in the Conclusion would be better suited for the Discussion. Reserve the Conclusion section of a brief summary and broader applicability.
References
Minor inconsistencies in formatting.
Author Response
Review 1
We thank the reviewer for all the comments. All the revisions have been underlined along the manuscript, except for the deleted unnecessary text.
The contribute of cognitive factors to understand compulsive buying behavior: Insights from shopping habits changes during Covid19 pandemic reports on participants’ online shopping (compulsive buying) behaviors prior to and during the COVID-19 pandemic in conjunction with individuals’ decision-making styles and working memory characteristics. This is an interesting study with applicability beyond pandemic-associated (highly stressful time) outcomes. There are, however, a significant number of issues that must be addressed before publication.
General:
Please be consistent in the use and spelling of COVID-19. Covid19, Covid, Covid19 pandemic, and coronavirus pandemic are some examples from the manuscript.
Re: We have chosen to use the term COVID-19 consistently along the manuscript
Grammar (e.g. word order, verb tense shifts, pluralization) needs considerable attention. I provide a few examples from my discussion of the various sections of the paper below, but they are not an exhaustive list.
Re: Thank you for your observation. The English grammar of manuscript has been revised by a native speaker in order to fix grammar mistakes.
The title should be revised: The contribution of cognitive factors to compulsive buying behavior: Insights… or something similar.
Re: the title has been revised according to the suggestion
Introduction
The first sentence is not needed and the claim of “exponential” growth is not supported by citation. Instead, begin During the COVID-19 pandemic, global e-commerce grew….
Re: the change has been done
Lines 39-43 – This sentence is difficult to follow.
Re: We have rephrased the sentence
Line 45 – do you mean social interactions?
Re: We substituted relationships with interactions as suggested
Line 46- instead of involved perhaps use created, led to, etc.
Re: As suggested, we substituted involved with led to
Line 48 – Restate the hypothesis. Perhaps “The present study aimed to assess changes in individuals’ online shopping behaviors prior to and during pandemic restrictions.”
Re: We modified the sentence according to your suggestion. see line 55-56
Line 56 – what is meant by shopping specific samples?
Re: We meant “customers of a shopping mall”, we have modified the sentence accordingly
Line 71 – buying/shopping or buying and shopping behaviour
Re: we changed the sentence accordingly
Line 87 – by internal factors do you mean affective and cognitive components? If so, please state that way.
Re: We modified the sentence, by adding that we refer to the affective and cognitive components, as suggested. see line 86-87
Line 92 – Wouldn’t employee friendliness be considered external factor?
Re: We deleted employee and we left ‘friendliness’ meaning a trait of personality similar to amicality. See line 95
Lines 106-108- move the first sentence to the end of the paragraph above.
Re: The change has been made. See line 99-100
Line 156 – A spontaneous decision-making style will be associated with a greater tendency toward CB, or Individuals characterized by a spontaneous decision-making style will show a greater tendency toward CB.
Re: The sentence has been modified accordingly the suggestion. see line 168
Materials and Methods
2.1 This section needs to be reorganized. Move the number of participants (Our sample was composed…) to the beginning of the paragraph. Include the power calculation information after the age range information. The descriptive information would be easier to follow in a table. It would also be good to see the data separately for males and females.
Re: We reorganized the paragraph, putting at the beginning the number of participants and later the power calculation information, and added a table about demographics, following reviewer’s suggestion. see line 170-178
2.2 Proofread carefully and condense. There are details for these instruments that are repeated. Additionally, tense shifts are particularly problematic in this section.
Re: Tense shift of these section were revised through the professional grammar revision of English language and unnecessary details have been deleted. See line 211-275
Line 247 – what is meant by Scores are calculated for each participant with a specific equation? The sentence following this sentence also does not make sense.
Re: We meant that to calculate the score of the scale a specific calculation [(Scoring equation = -9.69 + (Q1 x .33) + (Q2a x .34) + (Q2b x .50) + (Q2c x .47) + (Q2d x .33) + (Q2e x .38) + (Q2f x .31)], provided by the authors, should be employed. The calculation requires to add all the individual scores obtained in each question, and to subtract 9.69 points to the total score. Therefore, the higher is the negative score obtained, the higher is the tendency to compulsive buying. We clarified better this point in the manuscript. See line 250-253
Line 261 – sentence fragment
Re: the sentence fragment has been deleted.
Results
This first section should be referred to as Statistical Analyses and may be better suited to the previous section (Materials and Methods). Either use a subheading as you do in Section 3.1 or restate your variables instead of referring to H1, H2, etc. For example, In order to assess changes in individuals’ online shopping habits before and during the pandemic, series of repeated measures ANOVAs (although I am unclear why a paired-samples T-test wasn’t used since there are only 2 time points) were performed. Another possibility is to state your statistical approach more broadly, i.e. hierarchical regression was used to assess the contributions of demographic characteristics, depression, decision-making styles, and working memory on online shopping behaviors, etc. There is also a lot of repetition in the models (both here and in the actual presentation of the results). Perhaps a table would be better?
Re: As suggested by the reviewer, we reorganize the section by introducing a ‘statistical analyses section’ that introduces the kind of analyses performed (ANOVAs and Hierarchical regression); then we introduced tables to make description of results easier. See line 283-297
If you move the Statistical Analyses to the Methods section, then the Results section should begin with a statement about the overall purpose of the paper. You can then work through the results for each of the hypotheses. Again, consider presenting the data from the regression models in a table. There is so much repetition in this section.
Re: as suggested, we introduced a statement about the overall purpose of the paper and we presented data from the regression models within tables, deleting the repetitions along the text. see Line 294-297, 319, 323, 329, 334.
Discussion
This section needs to be carefully proofread. Some sentences (such as lines 432-437) ramble and are difficult to follow. Additionally, the discussion of nudging in the Conclusion would be better suited for the Discussion. Reserve the Conclusion section of a brief summary and broader applicability.
Re: We thank Reviewer for his/her suggestions; we divided this section between discussion and conclusion and revised it in order to improve the manuscript. See page 10-11
References
Minor inconsistencies in formatting.
Re: they have been fixed
The English language of manuscript has been revised by a native speaker. We attached the certification.

Reviewer 2 Report
This paper made an attempt to to explore the role of working memory and decision-making style in compulsive behaviour. A sample of 105 participants (65 F, 40 M) is collected in 2020. Results showed that during the Covid19 pandemic people spent much more time shopping online, made more purchases and invested more money. Moreover, both higher working memory deficits and spontaneous decision-making style predicted a greater tendency to compulsive buying. These results suggest the need to develop specific trainings in order to improve cognitive aspects related to shopping compulsive. Some revisions and clarifictions are needed to fully appreciate the manuscript.
It is suggested to divide the introduction into two parts: introduction and review. The current version put too many review contents in the introduction, which make the readers cannot get the main point directly at the beginning. Focus on background and meaning as well as brief summary about the contents in the introduction. The existing gaps could be given in the review section.
While I appreciate the vast literature reported, its discussion is not well organized. At the moment, it looks like an unstructured description of current papers, listed just one after the other. It needs to be reorganized around the specific objective of this paper with the aim to clarify the choices made in this paper and why these are relevant.
“Indeed, to prevent the pandemic’s spread, millions of people stopped their daily activities or moved them into smart working, avoided social relationships and were isolated at home (Anwari et al. 2021; Yao et al. 2022).” The statement needs evidence and supports. The two references are appropriate for supporting the statement.
Anwari, Nafis, Md Tawkir Ahmed, Md Rakibul Islam, Md Hadiuzzaman, and Shohel Amin. "Exploring the travel behavior changes caused by the COVID-19 crisis: A case study for a developing country." Transportation Research Interdisciplinary Perspectives 9 (2021): 100334.
Yao, W., Yu, J., Yang, Y., Chen, N., Jin, S., Hu, Y., and Bai, C., 2022. Understanding travel behavior adjustment under COVID-19. Communications in Transportation Research, 2, 100068. doi:10.1016/j.commtr.2022.100068.
Section 2.1 is suggested to use a table for summarizing the demographic statistics, which are more reader-friendly.
You used 5-point Likert for surveying. Did you conduct standardization before analysis?
In the discussion sections, there are large variations in the behaviors among different people (Xu et al. 2021; Ortúzar 2021). How do you consider this issue? Corresponding clarifications should be added. If the behaviroal heterogeneity is not considered, at least, a corresponding limitation statement should be provided to clarify this by adding the references.
A. Stathopoulos, S. Hess, Revisiting reference point formation, gains–losses asymmetry and nonlinear sensitivities with an emphasis on attribute specific treatment
Ortúzar, J. de D. (2021) ‘Future transportation: Sustainability, complexity and individualization of choices’
The collected sample size is not large and contains about 100 respondents. How can you justify the findings are reliable and convincing if the sample is not enough? Please add some clarifications.
Limitation statements should be added in a more detailed way after conclusions as many aspects need to be improved. For instance, is the finding general? How can further research improve present outocomes? An interesting future study direction is to use some quantitative behavioral models to further investigate the topic based on the collected data, such as hybrid choice models (Gao et al. 2020) and prospect theory (Gao et al. 2021; Li and Hensher et al. 2018). It is recommended that the future study point is added in the discussions.
Gao et al. . "Revealing psychological inertia in mode shift behavior and its quantitative influences on commuting trips." Transportation research part F: traffic psychology and behaviour 71 (2020): 272-287.
Z. Li, D. Hensher, Prospect theoretic contributions in understanding traveller behaviour: a review and some comments
Gao, K., Yang, Y. and Qu, X. (2021) Diverging effects of subjective prospect values of uncertain time and money
The writing should be carefully checked and revised in terms of grammar, equations, and expressions.
Author Response
Review 2
We thank the reviewer for all the comments. All the revisions have been underlined along the manuscript, except for the deleted unnecessary text.
This paper made an attempt to explore the role of working memory and decision-making style in compulsive behaviour. A sample of 105 participants (65 F, 40 M) is collected in 2020. Results showed that during the Covid19 pandemic people spent much more time shopping online, made more purchases and invested more money. Moreover, both higher working memory deficits and spontaneous decision-making style predicted a greater tendency to compulsive buying. These results suggest the need to develop specific trainings in order to improve cognitive aspects related to shopping compulsive. Some revisions and clarifictions are needed to fully appreciate the manuscript.
It is suggested to divide the introduction into two parts: introduction and review. The current version put too many review contents in the introduction, which make the readers cannot get the main point directly at the beginning. Focus on background and meaning as well as brief summary about the contents in the introduction. The existing gaps could be given in the review section.
Re: Thank you for the suggestion; accordingly, we divided the Introduction in two different sections (Background and Review): in the background section we focused on the meaning of compulsive buying, whereas in the review section we focused on prior literature and existing gaps on the topic.
While I appreciate the vast literature reported, its discussion is not well organized. At the moment, it looks like an unstructured description of current papers, listed just one after the other. It needs to be reorganized around the specific objective of this paper with the aim to clarify the choices made in this paper and why these are relevant.
Re: We thank Reviewer for his/her suggestion. We now argued the literature underling more which are our aims and why they are relevant. E.g. line 98-99, 107-111, 132-135, 137-143, 150-153
“Indeed, to prevent the pandemic’s spread, millions of people stopped their daily activities or moved them into smart working, avoided social relationships and were isolated at home (Anwari et al. 2021; Yao et al. 2022).” The statement needs evidence and supports. The two references are appropriate for supporting the statement.
Anwari, Nafis, Md Tawkir Ahmed, Md Rakibul Islam, Md Hadiuzzaman, and Shohel Amin. "Exploring the travel behavior changes caused by the COVID-19 crisis: A case study for a developing country." Transportation Research Interdisciplinary Perspectives 9 (2021): 100334.
Yao, W., Yu, J., Yang, Y., Chen, N., Jin, S., Hu, Y., and Bai, C., 2022. Understanding travel behavior adjustment under COVID-19. Communications in Transportation Research, 2, 100068. doi:10.1016/j.commtr.2022.100068.
Re: Thank you, we included in the text the suggested references. See line 43
Section 2.1 is suggested to use a table for summarizing the demographic statistics, which are more reader-friendly.
Re: A table including demographic statistics has been added. See line 194
You used 5-point Likert for surveying. Did you conduct standardization before analysis?
Re: No, we did not. The questions in table 2 investigate purchasing habits and they not provide a cut-off on a pathological condition, so we believe they do not require standardisation. They are comparable to demographic questions. Different is the case concerning the questionnaires used to measure the presence of depression, compulsive buying, working-memory deficit and its components and decision-making styles, which are instead standardised because providing a pathological or a predispositional profile of the individual.
In the discussion sections, there are large variations in the behaviors among different people (Xu et al. 2021; Ortúzar 2021). How do you consider this issue? Corresponding clarifications should be added. If the behaviroal heterogeneity is not considered, at least, a corresponding limitation statement should be provided to clarify this by adding the references.
- Stathopoulos, S. Hess, Revisiting reference point formation, gains–losses asymmetry and nonlinear sensitivities with an emphasis on attribute specific treatment
Ortúzar, J. de D. (2021) ‘Future transportation: Sustainability, complexity and individualization of choices’
Re: As suggested by the Reviewer, we introduced a limitation statement concerning the large variation in human behaviours, quoting Xu et al. 2021; Ortúzar 2021. See line 465-470
The collected sample size is not large and contains about 100 respondents. How can you justify the findings are reliable and convincing if the sample is not enough? Please add some clarifications.
Re: In order to calculate the sample size we used the G*power analysis. In the sample section is specified “To determine the sample size, a power calculation was performed using GPower 3.1. [58]. To run a multiple regression analysis (considering two predictors: working memory and spontaneous decision-making style) the following parameters were used: effect sizef2 = .15 - medium magnitude; alpha = .05; power = .90. This gave a suggested sample size of at least 88 participants”. However, we introduce the limitation concerning the large variation in human behaviour as written before. See line 465-470
Limitation statements should be added in a more detailed way after conclusions as many aspects need to be improved. For instance, is the finding general? How can further research improve present outcomes? An interesting future study direction is to use some quantitative behavioral models to further investigate the topic based on the collected data, such as hybrid choice models (Gao et al. 2020) and prospect theory (Gao et al. 2021; Li and Hensher et al. 2018). It is recommended that the future study point is added in the discussions.
Gao et al. . "Revealing psychological inertia in mode shift behavior and its quantitative influences on commuting trips." Transportation research part F: traffic psychology and behaviour 71 (2020): 272-287.
- Li, D. Hensher, Prospect theoretic contributions in understanding traveller behaviour: a review and some comments
Gao, K., Yang, Y. and Qu, X. (2021) Diverging effects of subjective prospect values of uncertain time and money
Re: As suggested by the Reviewer, we introduce the issues in limits and future research. See line 454-458
The writing should be carefully checked and revised in terms of grammar, equations, and expressions.
Re: The English language of manuscript has been revised by a native speaker. We attached the certification.

Round 2
Reviewer 2 Report
Thanks for the authors addressing my comments.